# Adalimumab in Treating Refractory Livedoid Vasculopathy

**DOI:** 10.3390/vaccines10040549

**Published:** 2022-04-01

**Authors:** Xiao-Wen Huang, Huan-Xin Zheng, Meng-Lei Wang, Wan-Mei He, Mei-Xin Feng, Kang Zeng, Li Li

**Affiliations:** Department of Dermatology, Nanfang Hospital, Southern Medical University, Guangzhou 510150, China; huangxw@smu.edu.cn (X.-W.H.); xin962@sina.com (H.-X.Z.); menglei@smu.edu.cn (M.-L.W.); hwm0904@163.com (W.-M.H.); fengmeixin970208@163.com (M.-X.F.); nfpfkzk@126.com (K.Z.)

**Keywords:** livedoid vasculopathy, adalimumab, anti-TNF-α agent

## Abstract

Livedoid vasculopathy is a chronic, recurrent skin disorder. It seriously affects the quality of patients’ life. However, the pathogenesis has not been fully identified yet. Here, this retrospective study describes the successful use of anti-TNF-α agent adalimumab in three cases of refractory livedoid vasculopathy, which has not been reported previously. In addition, we provide some clinical evidence that adalimumab therapy is efficient in improving skin lesions and relieving the pain of livedoid vasculopathy.

## 1. Introduction

Livedoid vasculopathy (LV) is a rare cutaneous disease with manifestations of livedo reticularis, leg ulcerations, and atrophie blanche. Lesions of LV usually distribute symmetrically on the dorsal feet, ankles, and lower extremities. The ulceration results in atrophic, porcelain-white scars termed atrophie blanche [1]. Patients with LV always feel pain and hyperesthesia surrounding the affected area, causing a severely reduced quality of life.

The etiology and pathogenesis of LV have remained indefinite until now; however, various etiopathological factors may be involved, including microvascular thrombosis and endothelial proliferation. Autoimmune reaction may also be contributory [2]. To date, thromboembolic vs. inflammatory pathophysiology has long-lasting ambiguity [3,4]. A recent review revealed the potential etiological role of plasminogen activator inhibitor (PAI)-1 in the pathogenesis of LV [5]. PAI-1 is an inhibitor of the endogenous fibrinolytic system in blood coagulation and has gradually drawn attention because of its potential role in the pathogenesis and mechanism of LV. It has been reported inflammatory cytokines, such as tumor necrosis factor-α (TNF-α), regulate the expression of genes impairing the function of endothelial cells and inducing the production of PAI-1 [6]. So, TNF-α may be involved in the occurrence of LV.

Nowadays, the treatments for LV are challenging as there are no practical therapeutic guidelines yet. The conventional therapies include anticoagulants, anti-platelets, fibrinolytic, vasodilators, anti-inflammatory agents, immunosuppressants, and supportive measures [7]. In some cases, it is necessary to perform the combination therapy. However, the differences in treatment of LV between individuals are significant, and it still lacks solutions to satisfy the recurrence. Recently, several LV cases showed that anti-TNF-α agents and Janus-activated kinase (JAK) inhibitors might have a positive therapeutic response, typically for refractory cases [8,9]. Herein, this retrospective study described three refractory LV patients with poor responses to traditional therapeutics who were successfully treated with the anti-TNF-α agent adalimumab. Moreover, it is evident that adalimumab treatment promotes wound healing in LV patients.

## 2. Case Series Reports

Patient 1 was a 25-year-old female worker with an 11-year history of non-healing ulcers over the distal lower extremities. She complained of gradually expanding ulceration and intermittent pain. Physical examination revealed porcelain-white stellate scars with surrounding hyperpigmentation of both her legs. The solitary ulcer was covered with black crusts on the dorsum of the feet, as shown in Figure 1 (left). Patient 2 was a 19-year-old female student who presented with a reticulate pattern of ecchymosis and impending ulceration on both her feet (Figure 2, above). She complained of a three-year recurrence and gradually worsening pain. Patient 3 was a 26-year-old female freelancer. At presentation, she gave a history of developed stellate ulcers with marked erythema on the bilateral ankles eight years ago. Violaceous streaks were located above the ankle, with multiple unregular ulcerations separating on the ankle (Figure 3, left).

Skin biopsy specimens were stained with hematoxylin-eosin for histopathologic evaluation. As shown in Figure 4, intraluminal hyaline thrombi (arrowhead) and fibrinoid necrosis in the blood vessel wall (asterisks) were observed in the skin biopsies from all three patients. They were absent from perivascular neutrophil infiltrates and leukocytoclasia. In lab tests, C-reactive protein, erythrocyte sedimentation rate, coagulation function, anti-nuclear antibody, anti-cardiolipin antibodies, protein-C, protein-S, anti-thrombin-III, and homo-cysteine were normal or negative in patients 1 and 2. In contrast, patient 3 had an increased erythrocyte sedimentation rate (Table 1). These three patients were all young females with a long history of recurrent ulcerations involving the lower extremities. The clinical manifestations of cases here are pretty characteristic. Therefore, clinical and histopathological diagnoses of livedoid vasculopathy were made for these three patients.

Of note, all three patients provided a history of treatment failure of anabolic steroids and immunosuppressive therapy, such as methotrexate and thalidomide in the past. Patient 1 had taken methylprednisolone for half a year, with a maximum dose of 40 mg per day. She had also received irregular treatment with methotrexate without providing a specific dose and duration by herself. Patients 2 and 3 had received prednisolone irregularly for 4 months and 7 months, respectively. However, they mentioned the occasional improvement while on medication. Patient 1 had also received the anti-platelet agent dipyridamole without any long-term remission. Patient 3 had used anticoagulants rivaroxaban 10 mg twice daily for 2 months before here. She complained of severe pain and the urgent need to relieve the pain.

We discontinued the previous medicines, such as anabolic steroids and immunosuppressive agents. Considering the recurrence of ulcers and the complaint of pain, which severely affected the daily life of patients 1 and 2, we started the treatment with the anti-TNF-α agent adalimumab. The specific therapy dosage was a subcutaneous injection of adalimumab 40 mg every 2 weeks for 16 weeks. It turned out that skin lesions improved with remarkable pain relief in patients 1 and 2 by the fourth month of treatment, only leaving some unregular scars and pigmentation (Figure 1, right and Figure 2, bottom).

Although patient 3 showed some remission of lesions on anticoagulant therapy, due to the intermittent pain episodes and the patient’s request to change treatments, we suggested adalimumab as her monotherapy. After 6 months of adalimumab, patient 3 showed significant remission of ulcers and pain relief (Figure 3, right). Follow-up was paramount. The longest follow-up period for these three patients was more than 2 years, and no recurrence has been observed. All patients feel satisfied with the therapeutic effect of adalimumab.

## 3. Discussion

Chronic ulcers are often associated with different severity of infections and pain, which have certain limitations on patients’ daily activities. Most chronic leg ulcers are due to venous disease, differentiated from wounds of the foot. However, the aims of management still are wound healing and preventing secondary infections. To manage these ulcerations effectively, clinicians must make accurate diagnoses and manage the etiology, yet, underlying causes present a particular challenge for clinicians. Diverse factors may be involved, such as venous disease, arterial insufficiency, diabetic neuropathy, pressure, lymphedema, infection, calciphylaxis, drug induction, vasculitis, and autoimmune disease [10]. In some cases, multiple factors are confounded, making the treatment more difficult. Nevertheless, clinicians can follow some regular processes when treating patients with ulcers. Obtaining a detailed inquiry of the onset, development and previous treatment modalities is the initial step in assessing chronic leg ulcerations. Then, perform a meticulous physical examination, including noting the skin surface and checking the vein’s condition. Ulcers of LV presented here are irregular and incurable. As predicted in the literature, lesions of LV are characterized by small irregular ulcers, usually developed from erythematous or purpuric papules. These ulcers become crusted and heal progressively; then, there is the appearance of fresh ulcers that eventually heal to form white atrophic scars. The presentation of ulcers here is shallow and irregularly shaped, primarily affecting the gaiter region and malleolus, which are the atypical location of venous ulcers.

Cases of LV here are from young women. The incidence of LV is three times more common in females than in males, especially in patients aged 15 to 50 years [11]. Some risk factors of chronic leg ulcers can be excluded among young females, such as diabetes, smoking, and cardiovascular comorbidities. Defining risk factors and informing patients are prerequisites in the treatment. LV is commonly associated with vascular pathology. Identifying the involved pathophysiological element of lesions and differentiating it from other lower extremities ulcers is critical for effective treatments. Although the standard guideline for treatment is lacking, the primary goal of therapy is ulceration healing and pain reduction. We know that anticoagulants are the most commonly reported effective category for LV, and anabolic steroids are the second most frequently used [12]. However, failure of improvement on corticosteroid therapy happened in these three patients here. According to the aims of ulceration management, which are to correct the underlying cause of the ulceration and achieve wound healing using the most current guideline or available cases reports, we tried to apply the anti-TNF-α agent as the monotherapy to patients here. These refractory cases of LV have a good response to Adalimumab, which supplements the existing literature. We confirmed that adalimumab helps achieve treatment goals of improving skin lesions, alleviating pain, and preventing recurrence efficacy.

TNF-α is a key regulator of innate immunity, playing an important role in regulating Th1 immune responses. Dysregulated TNF contributes to numerous pathological situations. Monoclonal antibodies targeting TNF-α have revolutionized the therapeutic approach to difficult-to-treat autoimmune and inflammatory diseases. The anti-TNF-α drugs have been reported as a valid alternative for treating venous events. More recent reports suggest the efficacy of TNF-α therapy in Behçet’s disease, a complex vasculitis characterized by superficial venous thrombus and deep vein thrombosis [13]. In the treatments of Behcet’s disease, corticosteroids are the mainstay, while colchicine, ciclosporin-A, cyclophosphamide, IFN-alpha, and other biologic agents are used as the induction and/or maintenance therapy. Similar to Behcet’s disease, endothelial cell plasminogen activation, platelet dysfunction, and fibrin accumulation are the recognized pathogenesis of LV. For limited reports of anti-TNF-α agents used in treating LV, we tried to figure out the related mechanism from other skin diseases that respond positively to anti-TNF-α agents, such as Behçet’s disease. One study of Behçet’s disease mentioned that vascular thrombosis is controlled with immunosuppressant drugs rather than anticoagulants because the inflammation promotes vascular events in Behçet’s syndrome. We speculate that adalimumab is effective in these three cases of LV here partly because the inflammation promotes the hypercoagulability state, which leads to cutaneous vessels intraluminal thrombosis.

Many physiological processes and activities are involved, including thrombosis, inflammation, fibrosis, wound healing, angiogenesis, cell migration, and adhesion [5]. The main mechanisms are hypercoagulability and inflammation. The contribution of inflammatory and coagulation components in the pathogenesis of Behçet’s syndrome vascular events are inspirational for understanding LV and choosing effective therapeutic strategies. Additionally, it is well known that patients with psoriasis obtain benefits from anti-TNF-α therapy, especially for patients with metabolic syndrome concomitantly [14]. The mechanism is partly associated with reducing endothelial activation by anti-TNF-α agents. Some research predicted that adalimumab opposed macrophage cholesterol accumulation, particularly relevant on macrophages expressing membrane TNF-α [15]. Endothelial cell damage is believed to be one cause of thrombus formation in the capillary vasculature, which is involved in the pathogenesis of LV. Focusing on patients with active ulcers, we find some inspiration from treating pyoderma gangrenosum (PG), an inflammatory skin disease that rapidly progresses. In a recent review [16], TNF-α inhibitors demonstrated significant effectiveness with a response and complete response rates supporting the use of TNF-α inhibitors to treat PG. Although the etiology of PG was unclear until now, T cell activation, abnormal neutrophil migration, and tumor necrosis factor appear to play significant roles [17]. Based on the above information, the activation of vascular endothelial cells and T cells may be the targets of adalimumab in treating LV, which still need further research to verify.

LV is confused with leukocytoclastic vasculitis (LCV) in some complicated situations, for the two diseases primarily manifest as chronic and painful ulcers on the lower extremities [18]. Regarding etiology, LV is a rare thrombo-occlusive disorder, while LCV is a common form of small vessel vasculitis. In recent years, it has also been reported that some cases of LV present as the second form related to some conditions, including systemic lupus erythematosus, scleroderma, and antiphospholipid antibody syndrome [19]. The inflammatory response is involved in the pathogenesis of LV and is primarily considered subsequent to the procoagulant. Clinically, LV manifests as painful erythematous lesions related to the inflammatory response, mediated by different factors, including cytokines such as TNF-α. The gradually comprehensive understanding of etiology points out more therapeutic options.

We also noticed publications mentioned anti-TNF-α therapy had been increasingly associated with drug-induced autoimmune diseases, such as cutaneous vasculitis. For example, a young woman with ulcerative colitis received infliximab for one year, Leukocytoclastic Vasculitis was induced [20]. A retrospective review summarized 213 cases of anti-TNF-α agents induced vasculitis and reported the presence of systemic vasculitis was in an equally large number [21]. However, it is also well known that some biologic agents successfully treat systemic vasculitis. From the cases here, anti-TNF-α agents are efficient for LV, one type of immune-related small vessel disease. Nevertheless, the role of anti-TNF-α agents in the treatment of vasculitis remains controversial. Our cases have some features in the outcome, showing complete recovery after taking the monotherapy of adalimumab. It can be partly explained by the potential role of individual genetic susceptibility. From this aspect, we speculate anti-TNF-α therapy not only regulates inflammation responses but also participates in the development of coagulation and thrombosis.

There is confusion about LV, such as whether the chronic and recurrent disease is thrombotic or inflammatory. Recently, studies suggest that vasculitis plays a vital role in the pathogenesis of LV [7,12]. The positive therapeutic response to immunosuppressive and immunomodulatory agents supports this view [8,9,22]. It has been reported that intravenous immunoglobulin may be appropriate for refractory LV, related to the modulation of immune responses. Herein, for stubborn cases treated with steroids and antiplatelet agents, we prescribed adalimumab and observed the efficacy was promising. To our knowledge, this is the first evaluation of adalimumab in treating LV, although etanercept has been shown effective in other publications. Five patients with livedoid vasculopathy who were resistant to steroids, antiplatelets, or danazol therapy were treated with the anti-TNF drug, as reported by Hong-zhong Jin in 2022. They chose etanercept 25–50 mg once a week for 12 consecutive weeks. It turned out that the disease severity in all patients significantly improved [8]. Among these three anti-TNF agents, etanercept, adalimumab, and infliximab, adalimumab is a fully human immunoglobulin G1 monoclonal antibody, with relatively low immunogenicity. Additionally, IV infliximab is not as convenient for patients as hypodermic injection of adalimumab. Considering the total cost, infliximab is also a beneficial choice. In addition, tofacitinib, the JAK inhibitor, has been reported could contribute to the healing of LV ulcers [9]. The three patients here are young females with a long history of LV. Two of them had received high doses of corticosteroids in conjunction with the immunosuppressive agent; however, ulceration is still recurrent. Another patient had a poor response to corticosteroids. Taking into consideration the adverse effects and resistance to steroids, adalimumab was suggested to use and seems to be promising. Regarding the recurrence, the follow-up is still occurring for the three patients here, and no rising lesions have been reported by them. For the limitation of anti-TNF agents used in treating LV, we are temporarily unable to provide more helpful information. That is why we consider the cases here as valuable. However, safety and efficacy need to be studied in a large population of patients.

We might ignore the poor wound healing of LV in some cases, such as the presentation of thick crusts in patient 1. Although crusted, the ulceration did not improve. After receiving adalimumab, it is easy to remove the crust, demonstrating advanced healing. According to the publication, Cao et al. reported adalimumab induced a wound healing profile in patients with hidradenitis suppurative. Given that macrophages play pivotal roles throughout the wound healing process [23], here we speculate that in LV, adalimumab might also improve wound healing.

In summary, although the pathophysiology of LV is incompletely understood, and there is no consensus on the optimal treatment modality of LV, data from the case series here confirm the efficiency of adalimumab in treating LV, which indicates adalimumab may be a promising strategy for treating refractory LV. Furthermore, the successful use of anti-inflammatory agents in LV also suggests an essential role of inflammation.

## Figures and Tables

**Figure 1 vaccines-10-00549-f001:**
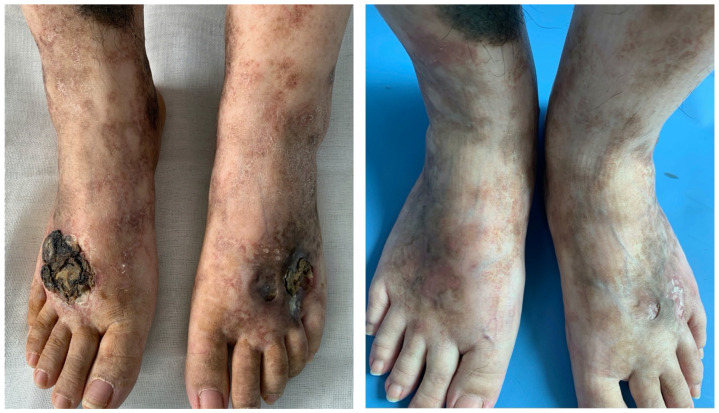
Clinical images of patient 1 before and 4 months after adalimumab treatment.

**Figure 2 vaccines-10-00549-f002:**
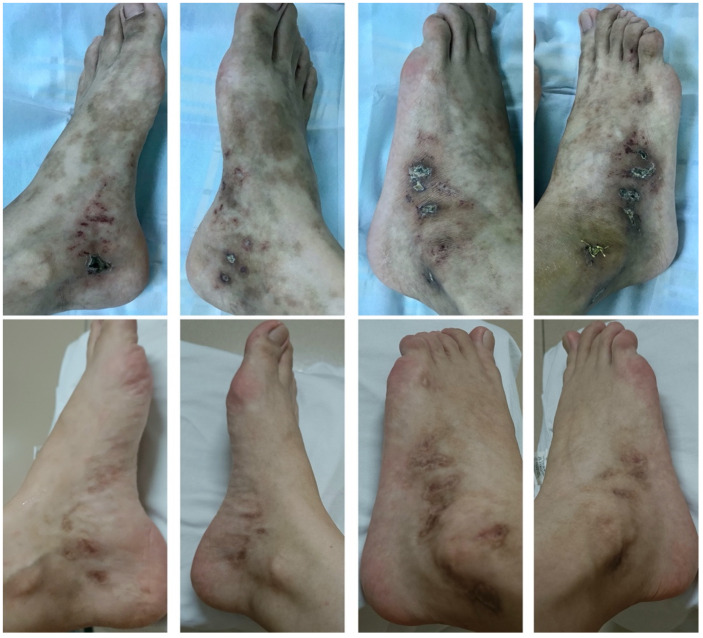
Clinical images of patient 2 before and 4 months after adalimumab treatment.

**Figure 3 vaccines-10-00549-f003:**
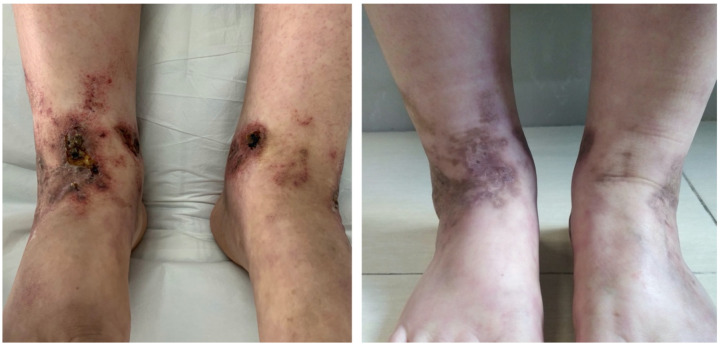
Clinical images of patient 3 before and 6 months after adalimumab treatment.

**Figure 4 vaccines-10-00549-f004:**
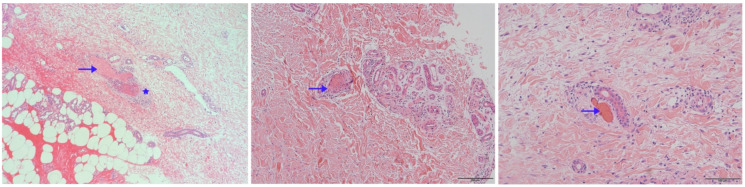
Skin biopsies of patients 1, 2 and 3.

**Table 1 vaccines-10-00549-t001:** The lab test results of the three patients.

		P1	P2	P3
**Age (y)**		25	19	26
**Medical History (y)**		11	3	8
**Lab Test**	Erythrocyte sedimentation rate (mm/h)	10	8	35
	C-reactive protein C (mg/L)	5.62	1.59	4.2
	Platelet count (109/L)	220	267	194
	Plasma prothrombin time (s)	10.6	7.9	7.5
	Activated partial thromboplastin time (s)	23.9	30	19.4
	D-Dimer (μg/mL)	0.33	0.17	0.53
	Protein-C	78%	70%	120%
	Protein-S	60%	55%	60%
	Anti-thrombin-III	80%	75%	110%
	Homo-cysteinem (μmol/L)	12.8	10.3	7.2
	Anti-nuclear antibody (u/mL)	15.31	5.88	18
	Anti-cardiolipin antibodies (PL/mL)	2.1	7.93	6.12

## Data Availability

Not applicable.

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
