# Peer review of "Adalimumab in Treating Refractory Livedoid Vasculopathy"

_vaccines, 2022, doi:10.3390/vaccines10040549_

Round 1

Reviewer 1 Report

The work is well-thought-out and well-organized. The brief introduction highlights problems related to the diagnosis and treatment of LV. The case reports are detailed and clearly show the course of existing treatment, which may be helpful for other centers. The discussion, based on the newest literature (the most references is from last 5 years), is extensive: it covers the diagnostics, differentiation and acceptable treatment of LV, and additionally it also characterizes the using of adalimumab in other diseases and the legitimacy of its application in LV. Photos of changes before and after treatment are very helpful in the perception of the work. 

Author Response

Thanks for your reviews and affirmations.

Reviewer 2 Report

In the paper "Adalimumab in treating refractory livedoid vasculopathy" Huag and coll. descibe three cases of livedoid vascuolopathy successufully treated by adalimumab

The paper beed wxtensive grammatical and language revision for improve readibility in addition is not clear what it's add to the existing literature 

Author Response

Thanks for your comments.

The manuscript has been revised by a native speaker and improved in readability. The existing literature mentioned anti-TNF-α agents might have a positive therapeutic response, typically for refractory cases, however, adalimumab has not been prescribed for LV previously. We tried to explain clearly the provoking thought by the three refractory cases of LV. From one aspect, adalimumab was used as the monotherapy on patients here and achieved a favorable response, which supplements the existing literature. From another aspect, the treatment's success inspires us to think about the mechanisms behind it, which is still in poor instructions. In the discussion part, it was emphasized that for the limitation of using adalimumab in treating LV, we recognize how it works on LV according to other vascular diseases, such as Behcet's disease. The supplements were marked in red in the manuscript (line 128-128, line 142-144). Please see the attachment. 

Reviewer 3 Report

Authors describe 3 patients successfully treated for livedoid vasculitis.  This type of cutaneous vasculitis is difficult to diagnose and treat. The evaluation of the livedoid vasculitis histological and the differential diagnostic examination are presented in detail together with therapeutic options. The part of discussion seeks to explain the different therapeutic responses of different type cutaneous vasculitis in each disease. By analysing the literature on this, it provides a useful approach to the treatment of the livedoid vasculitis.  

The English language use is good. The parts of the results and discussion are appropriate.

The article could use a table or a list of tasks laboratory tests to exclude other types of vasculitis interfering with the clinical picture

Author Response

Thanks for your reviews and affirmations. According to your advice, we have summarized the lab test results in a table and shown as Table 1 in the manuscript. Please see the attachment. 

Reviewer 4 Report

A description of the use of adalimumab in 3 patients with refractory livedoid vasculopathy. The topic is not related to the scope vaccines.

The study support the role of TNF in the pathogenesis of the disease.

The introduction is poor and does not describe mechanisms showing involvement of TNF in the pathogenesis of refractory livedoid vasculopathy. The authors mix etiology with mechanisms. A large part of data that should be in the introduction are given in the discussion section.

The description of cases is correct and nicely illustrated. It is unclear when improvement / remission (after how many doses) was achieved.

The discussion should be focused on comparison with the other anti-TNF drugs. It should be stated when and why the adalimumab should be used. Is it better than infliximab? or similarly effective ? If not why to use more expensive. The reader would like to know if there is a risk of recurrence after stopping  the anti-TNF treatment.

Author Response

Thanks for your comments. In the instruction, as you predicted, the pathogenesis and etiology of refractory livedoid vasculopathy were mixed together, we have revised the instruction, and marked in red in the manuscript (line 24-26). We also complemented some data of other anti-TNF drugs applied in treating LV. To our knowledge, five patients with livedoid vasculopathy who were resistant to steroids, antiplatelets, or danazol therapy were treated with the anti-TNF drug, as reported by Hong-zhong Jin in 2022 (J Dermatolog Treat. 2022 Feb;33(1):178-183). They chose etanercept 25-50 mg once a week for 12 consecutive weeks, and the disease severity in all patients significantly improved. Among the three anti-TNF agents, etanercept, adalimumab and infliximab, adalimumab is a fully human immunoglobulin G1 monoclonal antibody, with relatively low immunogenicity. additionally, IV infliximab is not so convenient for patients as hypodermic injection of adalimumab. considering the total cost, infliximab is also a beneficial choice. We added these data in the discussion part and marked in red (line 202-210, line 216-219). Regarding the recurrence, the follow-up is still going on for the three patients here, and no rising lesions have been reported by them. For the limitation of anti-TNF agents used in treating LV, temporarily unable to provide more effective information. That is why we consider the cases here are valuable.

Round 2

Reviewer 2 Report

The authors didn’t response adeguately to reviewer comments. 

Author Response

Thanks for your comments.

As mentioned, anti-TNF-α agent adalimumab has not been reported in treating LV in the previous literature, although anti-TNF-α agent etanercept showed significant improvement for refractory LV lesions (J Dermatolog Treat. 2022 Feb;33(1):178-183). We complemented these data in the discussion and marked in red (line 206-210), which provides physicians with some consideration when choosing adalimumab in LV treatment.

Reviewer 4 Report

The paper was improved somehow. Unfortunately the language (style) of new text is poor. 

An example: Adalimumab achieved a favorable response

Author Response

Thanks for your comments. We have revised the manuscript again and marked in red.

Round 3

Reviewer 2 Report

The manuscript has been improved and deserves publication 

Reviewer 4 Report

The English was improved. No further comments